# TLLC: Transfer Learning-based Label Completion for Crowdsourcing

**Wenjun Zhang** [1]   **Liangxiao Jiang** [1]   **Chaoqun Li** [2]

## Abstract

Label completion serves as a preprocessing approach to handling the sparse crowdsourced label matrix problem, significantly boosting the effectiveness of the downstream label aggregation. In recent advances, worker modeling has been proved to be a powerful strategy to further improve the performance of label completion. However, in real-world scenarios, workers typically annotate only a few instances, leading to insufficient worker modeling and thus limiting the improvement of label completion. To address this issue, we propose a novel transfer learning-based label completion (TLLC) method. Specifically, we first identify all high-confidence instances from the whole crowdsourced data as a source domain and use it to pretrain a Siamese network. The abundant annotated instances in the source domain provide essential knowledge for worker modeling. Then, we transfer the pretrained network to the target domain with the instances annotated by each worker separately, ensuring worker modeling captures unique characteristics of each worker. Finally, we leverage the new embeddings learned by the transferred network to complete each worker's missing labels. Extensive experiments on several widely used real-world datasets demonstrate the effectiveness of TLLC. Our codes and datasets are available at https://github.com/jiangliangxiao/TLLC.

## 1. Introduction

Supervised learning has achieved remarkable performance across diverse tasks, and its success relies on large-scale annotated data (Jiang et al., 2019; Zhang et al., 2023a). However, acquiring large-scale accurately annotated data from domain experts is often expensive and time-consuming (Lu et al., 2023). Fortunately, crowdsourcing offers a faster and more cost-effective alternative by employing crowd workers for annotation (Li et al., 2021). Due to varying expertise among workers, the labels collected from crowd workers contain a lot of noise (Xia et al., 2024). To address this, *repeated annotation* has been widely adopted, where each instance is annotated by multiple workers to obtain a multiple noisy label set (Sheng et al., 2008). Thus, simultaneously for multiple instances, a label matrix will be obtained. Subsequently, *label aggregation* is applied to infer the unknown true label of each instance based on this matrix.

To improve the performance of label aggregation, numerous methods have been proposed over the past decades (Dawid & Skene, 1979; Sheng et al., 2008; Zhang et al., 2016; Rodrigues & Pereira, 2018; Jiang et al., 2022; Li et al., 2023; Ying et al., 2024). These methods have gradually reached a consensus: when worker annotation is more accurate than random annotation, the more noisy labels an instance receives, the easier it becomes to infer its unknown true label (Chen et al., 2022; Zhang et al., 2023b). However, in real-world scenarios, each worker typically annotates only a small number of instances, and few labels are typically collected per instance to reduce cost, resulting in a highly sparse crowdsourced label matrix (Jung & Lease, 2012). This fact leads to label aggregation failing to achieve the expected performance relying solely on the existing labels in label matrix. To address this issue, *label completion* has been proposed to fill in more missing labels for the sparse label matrix, which is gaining increasing attention.

Although only a few methods have been proposed so far, they have already demonstrated that label completion can serve as a preprocessing approach to boost the effectiveness of the downstream label aggregation. Among them, recent advances further highlight the strength of worker modeling in improving the performance of label completion. Specifically, Yang et al. (2024) filter out potential noisy labels through worker modeling, and Wu et al. (2024) estimate worker similarity through worker modeling, both achieving notable improvements. However, to the best of our knowledge, despite its effectiveness, worker modeling is still constrained by the limited number of instances annotated by each worker. Insufficient annotated instances fail to accurately reflect the annotation ability of each worker,

[1]School of Computer Science, China University of Geosciences, Wuhan 430074, China [2]School of Mathematics and Physics, China University of Geosciences, Wuhan 430074, China. Correspondence to: Liangxiao Jiang <ljiang@cug.edu.cn>.

*Proceedings of the $42^{nd}$ International Conference on Machine Learning*, Vancouver, Canada. PMLR 267, 2025. Copyright 2025 by the author(s).

leading to insufficient worker modeling. Subsequently, insufficient worker modeling may misguide label completion, thereby limiting the improvement of label completion.

To address this issue, we propose a novel transfer learning-based label completion (TLLC) method. Specifically, we first identify all high-confidence instances from the whole crowdsourced data as a source domain and use it to pretrain a Siamese network. The abundant annotated instances in the source domain provide essential knowledge for worker modeling. Then, we transfer the pretrained network to the target domain with the instances annotated by each worker separately, ensuring worker modeling captures unique characteristics of each worker. Finally, we leverage the new embeddings learned by the transferred network to complete each worker's missing labels. In general, the contributions of this paper can be summarized as follows:

- We reveal the limitations of existing methods that leverage worker modeling to improve label completion. The fact that each worker annotates only a few instances leads to insufficient worker modeling and thus limits the improvement of label completion.

- We construct source and target domains for worker modeling using crowdsourced data. The source domain provides essential knowledge for worker modeling and target domains ensure worker modeling captures unique characteristics of each worker.

- We propose a novel transfer learning-based label completion (TLLC) method. TLLC introduces transfer learning to avoid insufficient worker modeling and leverages the new embeddings learned by the transferred network to complete missing labels.

The rest of this paper is organized as follows: Section 2 briefly reviews closely related work. Section 3 provides a detailed description of our proposed TLLC. Section 4 reports the experiments and results. Section 5 concludes this paper and outlines future research directions.

## 2. Related Work

In this section, we briefly review the related work on label completion and transfer learning.

**Label completion.** In crowdsourcing scenarios, label completion was initially proposed by Jung & Lease (2012). They applied probabilistic matrix factorization (PMF) to label completion, successfully completing missing labels in binary crowdsourcing scenarios. Inspired by collaborative filtering (Resnick et al., 1994), Watanabe & Kashima (2014) assumed that workers with similar annotation tendencies are more likely to assign the same labels. Subsequently, they

*Table 1.* Differences among existing label completion methods.

| Method | Label matrix | Instance attributes | Worker modeling | Applicable scenarios |
|---|---|---|---|---|
| Jung & Lease (2012) | ✓ | × | × | Binary |
| Watanabe & Kashima (2014) | ✓ | × | × | Binary |
| Zhou et al. (2016) | ✓ | × | × | Multi-class |
| Yang et al. (2024) | ✓ | ✓ | ✓ | Binary |
| Wu et al. (2024) | ✓ | ✓ | ✓ | Multi-class |

estimated worker similarities and leveraged these similarities to complete missing labels in binary crowdsourcing scenarios. Zhou & He (2016) used a three-dimensional tensor to represent the crowdsourced label matrix, and then performed tensor augmentation and completion to complete missing labels. Recently, worker modeling has been introduced into label completion, achieving significant progress. Yang et al. (2024) utilized worker modeling to improve the PMF-based label completion method (Jung & Lease, 2012). They modeled each worker by training a classifier and used the classifier to filter out potential noisy labels annotated by this worker before PMF. Wu et al. (2024) proposed a worker similarity-based label completion method called WSLC. WSLC first modeled each worker by learning a correlation vector between worker labels and instance attributes. Then, WSLC measured the cosine similarity between correlation vectors as worker similarity and used worker similarity to perform weighted voting for estimating missing labels. Table 1 summarizes the differences among the above methods, including whether they utilize the label matrix, whether they utilize instance attributes, whether they utilize worker modeling, and their applicable scenarios.

**Transfer learning.** To alleviate the insufficient worker modeling problem, we introduce transfer learning into label completion. Based on whether the source and target domains share the same attribute space, transfer learning can be divided into homogeneous transfer learning and heterogeneous transfer learning (Weiss et al., 2016). Homogeneous transfer learning (Yao & Doretto, 2010; Shi & Sha, 2012; Moustakas & Kolomvatsos, 2024) is applicable when the source and target domains have identical attribute spaces, while heterogeneous transfer learning (Sukhija, 2018; Bica & van der Schaar, 2022; Syu et al., 2025) is applicable when the source and target domains have different attribute spaces. In crowdsourcing scenarios, both the source and target domains should be derived from the same crowdsourced data, ensuring identical attribute spaces. Therefore, this work draws inspiration from homogeneous transfer learning to improve label completion.

## 3. The Proposed TLLC

### 3.1. Preliminary

Before introducing our proposed method in detail, we first define the basic notations for label aggregation and label completion in crowdsourcing. Let $D$ denote the crowd-

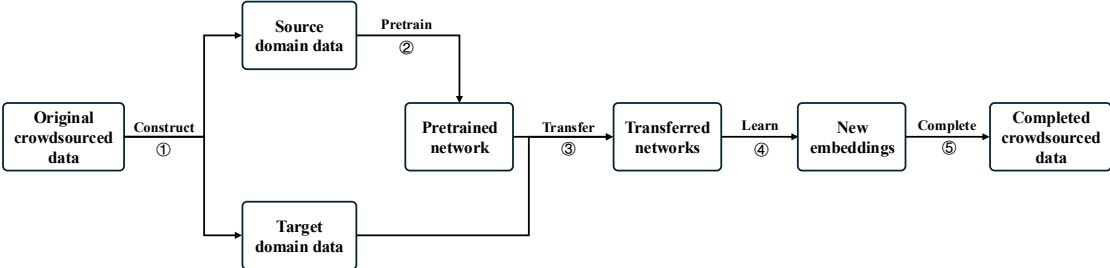

*Figure 1.* Overall framework of TLLC.

sourced data $\{(\boldsymbol{x}_i, \boldsymbol{L}_i)\}_{i=1}^N$, where $\boldsymbol{x}_i$ is the $i$-th instance in $D$, $\boldsymbol{L}_i$ is the multiple noisy label set of $\boldsymbol{x}_i$, and $N$ is the number of instances. $\boldsymbol{x}_i$ can be represented as $\{x_{i1}, \ldots, x_{im}, \ldots, x_{iM}\}$, where $M$ is the dimension of attributes, and $x_{im}$ is the attribute value of $\boldsymbol{x}_i$ on the $m$-th attribute $A_m$. $\boldsymbol{L}_i$ can be represented as $\{l_{ir}\}_{r=1}^R$, where $R$ is the number of workers, and $l_{ir}$ is the label of $\boldsymbol{x}_i$ annotated by the $r$-th worker $u_r$. $l_{ir}$ takes a value from a fixed set $\{-1, c_1, \ldots, c_q, \ldots, c_Q\}$, where $Q$ is the number of classes, $c_q$ is the $q$-th class, and $-1$ means that $u_r$ does not annotate $\boldsymbol{x}_i$. Based on these notations, we define label aggregation and label completion by **Definitions 3.1** and **3.2**.

**Definition 3.1.** Label aggregation infers the unknown true label $y_i$ of each instance $\boldsymbol{x}_i$ based on $\{(\boldsymbol{x}_i, \boldsymbol{L}_i)\}_{i=1}^N$, minimizing the error between the aggregated label $\hat{y}_i$ and the unknown true label $y_i$.

**Definition 3.2.** Label completion infers the missing label $l_{ir} = -1$ of each instance $\boldsymbol{x}_i$ based on $\{(\boldsymbol{x}_i, \boldsymbol{L}_i)\}_{i=1}^N$, ensuring that the completed label $\hat{l}_{ir}$ is the most likely label annotated to $\boldsymbol{x}_i$ by worker $u_r$.

In addition, we use $\boldsymbol{X}$ to represent all instances in $D$, $\boldsymbol{X}^r$ to represent the instances annotated by $u_r$, $\boldsymbol{L}^r$ to represent the labels $u_r$ annotated for $\boldsymbol{X}^r$, and $\bar{\boldsymbol{X}}^r$ to represent the instances not annotated by $u_r$. Meanwhile, to simplify the complexity of label aggregation and label completion, $D$ satisfies **Assumption 3.3** in this paper. For clarity, all notations defined in this paper are summarized in a table, which is provided in **Appendix A** due to the limited pages.

**Assumption 3.3.** The annotation difficulty of instances in $D$ is the same across all classes.

As discussed above, to alleviate the insufficient worker modeling problem, we propose a novel transfer learning-based label completion (TLLC) method. Its framework can be graphically shown in Figure 1. Firstly, we construct the source and target domain data from the original crowdsourced data. Secondly, we pretrain a Siamese network with the source domain data. Thirdly, we transfer the pretrained network with each worker's corresponding target domain data, respectively. Fourthly, we use the transferred network to learn new embeddings for each worker. Fifthly, we com-

plete each worker's missing labels with new embeddings to obtain a completed crowdsourced data.

To this end, TLLC needs to address three key issues: 1) How to construct the source and target domains from a given crowdsourced data? 2) How to perform worker modeling via transfer learning? 3) How to perform label completion? In the following subsections, we provide a detailed description of TLLC based on these three key issues.

### 3.2. Source and Target Domains Construction

First, we define the *domain* and *task* in transfer learning by **Definitions 3.4** and **3.5**, respectively.

**Definition 3.4.** A domain $\mathcal{D}$ consists of an attribute space $\mathcal{X}$ and a marginal probability distribution $P(\boldsymbol{X})$, i.e., $\mathcal{D} = \{\mathcal{X}, P(\boldsymbol{X})\}$.

**Definition 3.5.** A task $\mathcal{T}$ consists of a label space $\mathcal{Y}$ and an objective predictive function $f$, i.e., $\mathcal{T} = \{\mathcal{Y}, f\}$.

Then, we can denote the source domain and the target domain as $\mathcal{D}_S$ and $\mathcal{D}_T$, respectively. Since there is only $D$ available in crowdsourcing scenarios, we let $\mathcal{X}_S = \mathcal{X}_T = \mathcal{X} = \{A_1, \ldots, A_m, \ldots, A_M\}$ and $\mathcal{Y}_S = \mathcal{Y}_T = \mathcal{Y} = \{c_1, \ldots, c_q, \ldots, c_Q\}$ in this paper. To learn $f_S$, we need to construct the source domain data $D_S$ from $D$. Inspired by confident learning (Northcutt et al., 2021), for each instance $\boldsymbol{x}_i \in D$, we first obtain the initial aggregated label $\hat{y}_i$ and the corresponding confidence $P(\hat{y}_i|\boldsymbol{L}_i)$ as follows:

$$\hat{y}_i = \arg\max_{c_q \in \mathcal{Y}} P(c_q|\boldsymbol{L}_i), \tag{1}$$

$$P(c_q|\boldsymbol{L}_i) = \frac{\sum_{r=1}^R \delta(l_{ir}, c_q)}{\sum_{q=1}^Q \sum_{r=1}^R \delta(l_{ir}, c_q)}, \tag{2}$$

where $\delta(\cdot)$ is an indicator function that returns 1 if its two parameters are identical, and 0 if its two parameters are different. Subsequently, we calculate the average confidence $\mu_{c_q}$ for $c_q$ as follows:

$$\mu_{c_q} = \frac{\sum_{i=1}^N \delta(\hat{y}_i, c_q) P(\hat{y}_i|\boldsymbol{L}_i)}{\sum_{i=1}^N \delta(\hat{y}_i, c_q)}. \tag{3}$$

**Algorithm 1** Source and Target Domains Construction

**Require:** crowdsourced data $D$.
**Ensure:** source and target domain data: $D_S$, $\{D_T^r\}_{r=1}^R$.
 1: **for** $i = 1$ to $N$ **do**
 2:    Calculate $\hat{y}_i$ and $P(c_q|\boldsymbol{L}_i)$ by Equations (1) and (2);
 3: **end for**
 4: **for** $q = 1$ to $Q$ **do**
 5:    Calculate $\mu_{c_q}$ by Equation (3);
 6: **end for**
 7: Construct $\boldsymbol{X}_S$ by Equation (4);
 8: Construct $D_S$ by Equation (5);
 9: **for** $r = 1$ to $R$ **do**
10:    Construct $D_T^r$ by Equation (6);
11: **end for**
12: **return** $D_S$, $\{D_T^r\}_{r=1}^R$.

Next, we can get $\boldsymbol{X}_S$ as follows:

$$\boldsymbol{X}_S = \{\boldsymbol{x}_i | P(\hat{y}_i|\boldsymbol{L}_i) \geq \mu_{\hat{y}_i}, \text{for } i = 1, 2, \ldots, N\}. \quad (4)$$

Finally, we construct the source domain data $D_S$ as follows:

$$D_S = \{(X_{Si}, l_{Si}) \mid \text{for } i = 1, 2, \ldots, |\boldsymbol{X}_S|\}, \quad (5)$$

where $|\boldsymbol{X}_S|$ is the number of instances in $\boldsymbol{X}_S$, $X_{Si}$ is the $i$-th instance in $\boldsymbol{X}_S$, $l_{Si}$ equals to the initial aggregated label of $X_{Si}$. For the target domain $\mathcal{D}_T$, we construct a target domain data $D_T^r$ for each worker $u_r$ as follows:

$$D_T^r = \{(X_i^r, L_i^r) \mid \text{for } i = 1, 2, \ldots, |\boldsymbol{X}^r|\}. \quad (6)$$

Ultimately, $D_S$ contains abundant high-confidence annotated instances, which provide essential knowledge for worker modeling. $D_T^r$ contains all instances annotated by worker $u_r$, which reflect the unique characteristics of $u_r$.

The whole construction process of $D_S$ and $\{D_T^r\}_{r=1}^R$ in TLLC is shown in **Algorithm 1**. In **Algorithm 1**, lines 1-3 calculate the initial aggregated labels and their confidences with a time complexity of $O(NQR)$. Lines 4-6 calculate the average confidences with a time complexity of $O(NQ)$. Line 7 constructs $\boldsymbol{X}_S$, and line 8 constructs $D_S$, both with a time complexity of $O(N)$. Lines 9-11 construct the target domain data $\{D_T^r\}_{r=1}^R$ with a time complexity of $O(NR)$. Considering only the highest-order terms, the overall time complexity of **Algorithm 1** is $O(NQR)$.

**Theorem 3.6.** *Constructing $D_S$ based on Equation (5) can reduce the generalization error in transfer learning.*

*Proof.* According to Ben-David et al. (2010), the generalization error of transfer learning can be expressed as follows:

$$\epsilon_T \leq \epsilon_S + L^1(\mathcal{D}_S, \mathcal{D}_T) + \lambda, \quad (7)$$

where $\epsilon_S$ and $\epsilon_T$ are the errors in the source domain and target domain, respectively. $L^1(\mathcal{D}_S, \mathcal{D}_T)$ is the $L^1$ divergence

of $\mathcal{D}_S$ and $\mathcal{D}_T$. $\lambda$ reflects the difference between $f_S$ and $f_T$. According to **Assumption 3.3** and Equation (4), we can get $P(\boldsymbol{X}_S) = P(\boldsymbol{X})$. Therefore, Equation (5) will not change $L^1(\mathcal{D}_S, \mathcal{D}_T)$. Meanwhile, since $\lambda$ is only related to $f_S$ and $f_T$, Equation (5) will not change $\lambda$. Equation (4) filters out low-confidence instances for $\boldsymbol{X}_S$, which reduces the noise in $D_S$. This means that $\epsilon_S$ will be reduced, so the upper bound of $\epsilon_T$ will be reduced. □

### 3.3. Worker Modeling

After constructing $D_S$ and $\{D_T^r\}_{r=1}^R$, we perform worker modeling via transfer learning. According to different transfer strategies, homogeneous transfer learning can be divided into four classes: instance-based, attribute-based, parameter-based, and relational-based (Weiss et al., 2016). Inspired by fine-tuning (Guo et al., 2020), we leverage parameter-based transfer learning to perform worker modeling.

Specifically, we set up both $f_S$ and $f_T$ as Siamese networks with the same structure (Li et al., 2022). Let $\widetilde{D}$ to denote $D_S$ or $D_T$, $\widetilde{f}$ to denote $f_S$ or $f_T$, we first generate the training data $D'$ using $\widetilde{D}$ as follows:

$$D' = \{(\widetilde{X}_i, \widetilde{X}_j, y'_{ij}) \mid \text{for } i, j = 1, 2, \ldots, |\widetilde{\boldsymbol{X}}|\}, \quad (8)$$

where $\widetilde{\boldsymbol{X}}$ contains all instances in $\widetilde{D}$, $y'_{ij}$ is the supervisory information for training $\widetilde{f}$. Let $l_i$ denote the label corresponding to $\widetilde{X}_i$ in $\widetilde{D}$. We set $y'_{ij}$ to 0 if $l_i = l_j$, otherwise to 1. $\widetilde{f}$ has two parts, $\widetilde{f}_g$ and $\widetilde{f}_d$. $\widetilde{f}_g$ is used to learn a new embedding $\boldsymbol{z}$ for an instance $\boldsymbol{x}$ as follows:

$$\boldsymbol{z} = \widetilde{f}_g(\boldsymbol{x}) = \{z_1, \ldots, z_k, \ldots, z_K\}, \quad (9)$$

where $K$ is the dimension of the new embedding. $\widetilde{f}_d$ is used to calculate the Euclidean distance between new embeddings $\boldsymbol{z}_i$ and $\boldsymbol{z}_j$ as follows:

$$\widetilde{f}_d(\boldsymbol{z}_i, \boldsymbol{z}_j) = \sqrt{\sum_{k=1}^K (z_{ik}, z_{jk})^2}. \quad (10)$$

For each instance $(\boldsymbol{x}_i^1, \boldsymbol{x}_i^2, y'_i)$ in $D'$, we optimize $\widetilde{f}$ using the Mean Squared Error (MSE) loss function as follows:

$$\mathcal{L}_{mse} = \frac{1}{|\widetilde{\boldsymbol{X}}|^2} \sum_{i=1}^{|\widetilde{\boldsymbol{X}}|^2} (\widetilde{f}_d(\widetilde{f}_g(\boldsymbol{x}_i^1), \widetilde{f}_g(\boldsymbol{x}_i^2)) - y'_i)^2. \quad (11)$$

According to Equations (8) and (11), we first pretrain $f_S$ using $D_S$. After pretraining, $f_S$ has learned the essential knowledge for worker modeling, so we share its parameters with $f_T$. Then, we create a copy of $f_T$ as $f_T^r$ for $u_r$. Similarly, we fine-tune $f_T^r$ using $D_T^r$ by Equations (8) and (11). After fine-tuning, $f_T^r$ is transferred from $D_S$ to $D_T^r$, further capturing the unique characteristics of $u_r$. Therefore, fine-tuning $f_T^r$ is equivalent to modeling $u_r$.

---

**Algorithm 2** Worker Modeling

---

**Require:** source and target domain data: $D_S, \{D_T^r\}_{r=1}^R$.
**Ensure:** transferred networks $\{f_T^r\}_{r=1}^R$.
1: Generate data $D_S'$ using $D_S$ by Equation (8);
2: Pretrain $f_S$ using $D_S'$ by Equation (11);
3: Share the parameters of $f_S$ with $f_T$;
4: **for** $r = 1$ to $R$ **do**
5:    Copy $f_T$ as $f_T^r$;
6:    Generate data $D_T^{r'}$ using $D_T^r$ by Equation (8);
7:    Fine-tune $f_T^r$ using $D_T^{r'}$ by Equation (11);
8: **end for**
9: **return** $\{f_T^r\}_{r=1}^R$.

---

The whole process of worker modeling in TLLC is shown in **Algorithm 2**. In **Algorithm 2**, line 1 generates data $D_S'$ with a time complexity of $O(N^2)$. Let $O(g)$ represent the time complexity of $f_g$ when training $f$ (the time complexity of $f_d$ is $O(K)$). $O(g)$ depends on the scale of $f_g$ and is generally much larger than $O(K)$. Line 2 pretrains $f_S$ with a time complexity of $O(N^2 g)$. Line 3 shares the parameters of $f_S$ with $f_T$ and its time complexity of $O(g)$. Lines 4-8 fine-tune $\{f_T^r\}_{r=1}^R$ with a time complexity of $O(N^2 R g)$. Considering only the highest-order terms, the overall time complexity of **Algorithm 2** is $O(N^2 R g)$.

**Theorem 3.7.** *Parameter-based transfer learning can reduce the generalization error in worker modeling.*

*Proof.* $D_S$ and $D_T^r$ share the same attribute space. Meanwhile, both $f_T$ and $f_T^r$ are trained to map the attribute space to the label space. These consistencies make transfer learning feasible. Based on parameter-based transfer learning, we perform pre-training and fine-tuning with the same Siamese networks on $D_S$ and $D_T^r$, respectively. According to Equation (7), we adopt the same networks to reduce the difference between $f_T$ and $f_T^r$, thereby reducing $\lambda$, which further reduces the upper bound of $\epsilon_T$. This is equivalent to reducing the generalization error in modeling worker $u_r$. $\square$

**Theorem 3.8.** *When the noise in $D'$ follows an independent and identically distributed (i.i.d.) Gaussian distribution, worker modeling is robust to noise.*

*Proof.* We first use $y'$ and $y_t'$ to represent the supervisory information calculated using noisy labels and true labels, respectively. Then, $y'$ can be further expressed as follows:

$$y' = y_t' + \epsilon, \quad \epsilon \sim \mathcal{N}(0, \sigma^2). \tag{12}$$

Therefore, Equation (11) can be derived as follows:

$$
\begin{aligned}
\mathcal{L}_{mse} &= \mathbb{E}[(y' - d')^2] = \mathbb{E}[(y_t' + \epsilon - d')^2] \\
&= \mathbb{E}[(y_t' - d')^2] + 2\mathbb{E}[(y_t' - d')\epsilon] + \mathbb{E}[\epsilon^2]
\end{aligned} \tag{13}
$$

where $d'$ is the distance calculated by $f_d$. Since $\epsilon$ is independent of $y_t'$, $\mathbb{E}[\epsilon] = 0$, and $\mathbb{E}[\epsilon^2] = \sigma^2$, Equation (13) can finally be simplified to:

$$\mathcal{L}_{mse} = \mathbb{E}[(y_t' - d')^2] + \sigma^2 \tag{14}$$

Therefore, the effect of $\epsilon$ is a fixed constant, which means that worker modeling is robust to $\epsilon$. $\square$

### 3.4. Label Completion

From the defined Equations (8) and (11), it can be found that a transferred $f^r$ should satisfy:

$$f_d^r(\boldsymbol{z}_1^r, \boldsymbol{z}_2^r) < f_d^r(\boldsymbol{z}_1^r, \boldsymbol{z}_3^r), \text{ if } l_{1r} = l_{2r} \wedge l_{1r} \neq l_{3r}, \tag{15}$$

where $\boldsymbol{z}_1^r$ is $\boldsymbol{x}_1$'s new embedding learned by $f_g^r$, $l_{1r}$ is the label of $\boldsymbol{x}_1$ annotated by $u_r$. According to Equation (15), TLLC completes the missing labels of $u_r$ by calculating the distance between new embeddings of unannotated instances and annotated instances.

Specifically, we first use $f_T^r$ to learn the new embedding for each instance in $\boldsymbol{X}^r$ by Equation (9). Then, we calculate the centroid $\bar{\boldsymbol{z}}_q^r$ of new embeddings for each class $c_q$ as follows:

$$\bar{\boldsymbol{z}}_q^r = \{\bar{z}_{q1}^r, \ldots, \bar{z}_{qk}^r, \ldots, \bar{z}_{qK}^r\}, \tag{16}$$

where $\bar{z}_{qk}^r$ can be calculated as follows:

$$\bar{z}_{qk}^r = \frac{\sum_{i=1}^{|\boldsymbol{X}^r|} \delta(L_i^r, c_q) z_{ik}^r}{\sum_{i=1}^{|\boldsymbol{X}^r|} \delta(L_i^r, c_q)}. \tag{17}$$

Subsequently, we calculate the averaged distance $\bar{d}_q^r$ of new embeddings for each class $c_q$ as follows:

$$\bar{d}_q^r = \frac{\sum_{i=1}^{|\boldsymbol{X}^r|} \delta(L_i^r, c_q) f_d^r(\bar{\boldsymbol{z}}_q^r, \boldsymbol{z}_i^r)}{\sum_{i=1}^{|\boldsymbol{X}^r|} \delta(L_i^r, c_q)}. \tag{18}$$

Finally, for each unannotated instance $\bar{X}_i^r \in \bar{\boldsymbol{X}}^r$, we obtain its new embedding $\boldsymbol{z}_i^r$ by Equation (9) and complete $l_{ir}$ with $c_q$ if the following condition is satisfied:

$$f_{Td}^r(\boldsymbol{z}_i^r, \bar{\boldsymbol{z}}_q^r) \leq \bar{d}_q^r \wedge |\boldsymbol{X}^r| > 2Q. \tag{19}$$

Here, $f_{Td}^r(\boldsymbol{z}_i^r, \bar{\boldsymbol{z}}_q^r) \leq \bar{d}_q^r$ ensures that $\bar{X}_i^r$ is more similar to the instances annotated as $c_q$. $|\boldsymbol{X}^r| > 2Q$ encourages scenarios where $u_r$ annotates at least two instances for each class $c_q$, although the two are not strictly equivalent.

The whole process of label completion in TLLC is shown in **Algorithm 3**. In **Algorithm 3**, line 2 constructs $\boldsymbol{X}^r$ and $\bar{\boldsymbol{X}}^r$ for $u_r$ with a time complexity of $O(N)$. Lines 3-5 learn new embeddings for $\boldsymbol{X}^r$ with a time complexity of $O(Ng)$. Lines 6-9 calculate the centroid and the averaged distance for each class with a time complexity of $O(NQK)$. Lines 10-18 complete missing labels for $\bar{\boldsymbol{X}}^r$ with a time

**Algorithm 3** Label Completion

**Require:** crowdsourced data $D$, networks $\{f_T^r\}_{r=1}^R$.
**Ensure:** completed crowdsourced data $\hat{D}$.

1: **for** $r = 1$ to $R$ **do**
2:     Construct $\boldsymbol{X}^r$ and $\bar{\boldsymbol{X}}^r$ using $D$;
3:     **for** $i = 1$ to $|\boldsymbol{X}^r|$ **do**
4:         Learn $\boldsymbol{z}_i^r$ for $X_i^r$ by Equation (9);
5:     **end for**
6:     **for** $q = 1$ to $Q$ **do**
7:         Calculate $\bar{\boldsymbol{z}}_q^r$ for $c_q$ by Equation (16);
8:         Calculate $\bar{d}_q^r$ for $c_q$ by Equation (17);
9:     **end for**
10:    **for** $i = 1$ to $|\bar{\boldsymbol{X}}^r|$ **do**
11:       Learn $\boldsymbol{z}_i^r$ for $\bar{X}_i^r$ by Equation (9);
12:       **for** $q = 1$ to $Q$ **do**
13:          **if** Equation (19) holds **then**
14:            Complete a label $\hat{l}_{ir} = c_q$ for $\bar{\boldsymbol{X}}^r$;
15:            break;
16:          **end if**
17:       **end for**
18:    **end for**
19: **end for**
20: Reconstruct $\hat{D}$ with $\{\boldsymbol{X}^r\}_{r=1}^R$ and $\{\bar{\boldsymbol{X}}^r\}_{r=1}^R$;
21: **return** $\hat{D}$.

*Table 2.* Detailed network structure and parameter settings of $\widetilde{f}$.

| Layer type | Output dimension | Activation function |
|---|---|---|
| Input layer | 128 | ReLU |
| Fully connected layer | 64 | ReLU |
| Output layer | 2 | - |

complexity of $O(N(g + QK))$. Due to $O(g) >> O(QK)$, the time complexity of lines 1-19 should be $O(NRg)$. Line 20 reconstructs $\hat{D}$ with a time complexity of $O(NQR)$. Considering only the highest-order terms, the overall time complexity of **Algorithm 3** is $O(NRg)$.

By combining **Algorithms 1** to **3**, the overall time complexity of TLLC is $O(NQR + N^2Rg + NRg)$. Considering only the highest-order terms, the overall time complexity of TLLC is $O(N^2Rg)$, which is caused by worker modeling.

## 4. Experiments and Results

To validate the effectiveness of TLLC, we conduct extensive experiments. This section presents our experiments through three aspects: experimental setup, results, and analysis.

### 4.1. Experimental Setup

As shown in Table 1, the state-of-the-art WSLC (Wu et al., 2024) employs worker modeling and supports multi-class crowdsourcing scenarios, making it a key baseline for comparing with our proposed TLLC. We evaluate WSLC and TLLC by completing the same crowdsourced datasets and measuring the aggregation accuracy of label aggregation methods on their completed datasets, where aggregation accuracy represents the proportion of instances with aggregated labels matching true labels.

The label aggregation methods used in our experiments include majority voting (MV) (Sheng et al., 2008), ground truth inference using clustering (GTIC) (Zhang et al., 2016), differential evolution-based weighted soft majority voting (DEWSMV) (Tao et al., 2021), multiple noisy label distribution propagation (MNLDP) (Jiang et al., 2022), attribute augmentation-based label integration (AALI) (Zhang et al., 2023c), and label aggregation with graph neural networks (LAGNN) (Ying et al., 2024). For MV and GTIC, we use the existing implementations on the Crowd Environment and its Knowledge Analysis (CEKA) (Zhang et al., 2015) platform. For WSLC, DEWSMV, MNLDP, and AALI, we implement them on the CEKA platform. For LAGNN and TLLC, we implement them in Python. The parameter settings of all existing methods are consistent with those specified in their original papers. For TLLC, we set $K = 2$, the number of epochs to $Q$, and the batch size to 32. Additionally, we set the Siamese network $\widetilde{f}$ in TLLC to a small scale to ensure convergence, and the detailed network structure and parameter settings of $\widetilde{f}$ are shown in Table 2.

To provide a more comprehensive comparison, we conduct experiments on three different real-world datasets: *Income*, *Leaves*, and *Music_genre*. All three widely used datasets are collected through the online platform Amazon Mechanical Turk (AMT), and they represent different crowdsourcing requirements. Specifically, *Income* is collected from a binary scenario, while *Leaves* and *Music_genre* are collected from multi-class scenarios. In *Income* and *Leaves*, each instance is annotated by 10 workers, whereas in *Music_genre*, each instance is annotated by 4.2 workers. The proportion of missing labels in *Income*, *Leaves*, and *Music_genre* are 0.85, 0.88, and 0.90, respectively. Therefore, the label matrices of all three datasets are highly sparse, which aligns with the application scenarios of label completion. Due to the limited pages, more detailed information about these three datasets is provided in **Appendix B**.

### 4.2. Experimental Results

**Aggregation accuracy.** To reduce the impact of randomness on the experimental results, we independently repeat the experiments on each dataset ten times. Figure 2 shows the averaged aggregation accuracy of each label aggregation method after performing label completion by WSLC and TLLC, respectively. Based on these experimental results, we can summarize the following highlights:

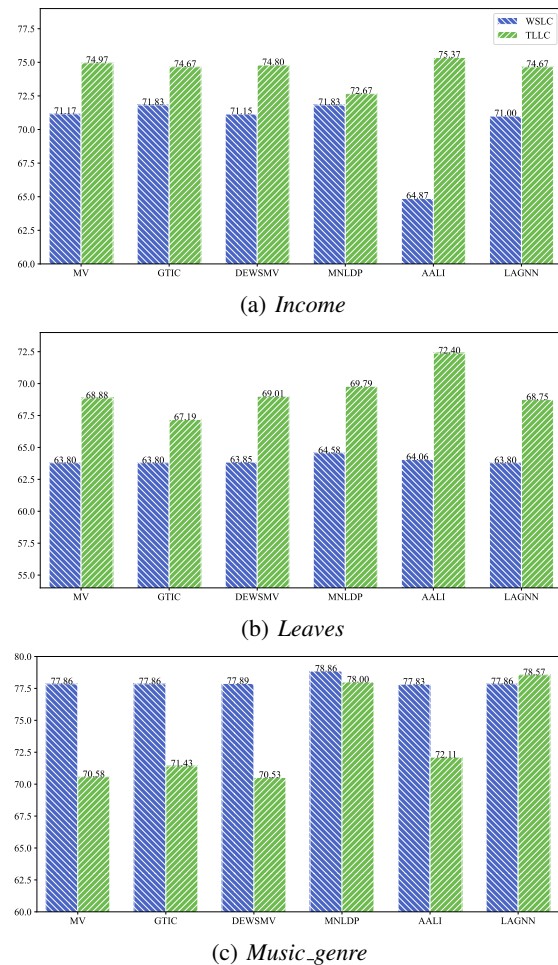

(a) *Income*

(b) *Leaves*

(c) *Music_genre*

*Figure 2.* Averaged aggregation accuracy (%) of each label aggregation method completed by WSLC and TLLC.

- On dataset *Income*, after completion by TLLC, the aggregation accuracy of each label aggregation method improves significantly. Specifically, the aggregation accuracies of MV (74.97%), GTIC (74.67%), DEWSMV (74.80%), MNLDP (72.67%), AALI (75.37%), and LAGNN (74.67%) after completion by TLLC are much higher than those of MV (71.17%), GTIC (71.83%), DEWSMV (71.15%), MNLDP (71.83%), AALI (64.87%), and LAGNN (71.00%) after completion by WSLC, respectively.

- On dataset *Leaves*, after completion by TLLC, the aggregation accuracy of each label aggregation method improves significantly. Specifically, the aggregation accuracies of MV (68.88%), GTIC (67.19%), DEWSMV (69.01%), MNLDP (69.79%), AALI (72.40%), and LAGNN (68.75%) after completion by TLLC are much higher than those of MV (63.80%), GTIC (63.80%), DEWSMV (63.85%), MNLDP (64.58%), AALI (64.06%), and LAGNN (63.80%) after completion by WSLC, respectively.

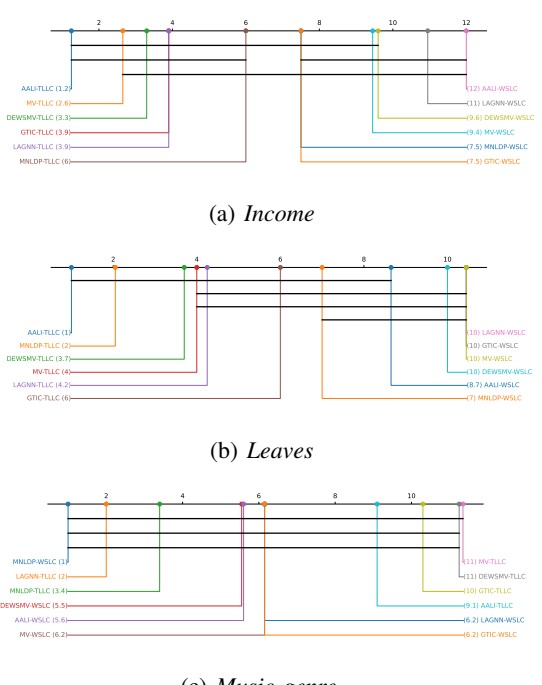

(a) *Income*

(b) *Leaves*

(c) *Music_genre*

*Figure 3.* Critical difference diagrams of significance tests.

- On dataset *Music_genre*, although TLLC does not reach its outstanding performance levels as in datasets *Income* and *Leaves*, it still maintains a relatively high upper bound of performance. Specifically, the aggregation accuracies of MNLDP (78.00%) and LAGNN (78.57%) after completion by TLLC are competitive with those of MNLDP (78.86%) and LAGNN (77.86%) after completion by WSLC, respectively.

**Significance tests.** In addition to comparing the averaged aggregation accuracies of ten repetitions, we directly perform a Friedman test with corresponding post-hoc tests (e.g., Nemenyi test) (Demsar, 2006; Jansen et al., 2023) on each dataset using the results of ten repetitions. These significance tests allow us to compare the performance differences between the label aggregation methods completed by WSLC and TLLC. Based on the test results, we present the critical difference (CD) diagrams in Figure 3. As shown in Figure 3, on datasets *Income* and *Leaves*, the label aggregation methods completed by TLLC achieve superior average rankings. Moreover, while TLLC's performance on dataset *Music_genre* is less pronounced compared to its performance on *Income* and *Leaves*, no statistically significant differences are observed between the label aggregation methods completed by WSLC and TLLC.

### 4.3. Discussion and Analysis

The experimental results above clearly validate the effectiveness of TLLC. Specifically, on datasets *Income* and

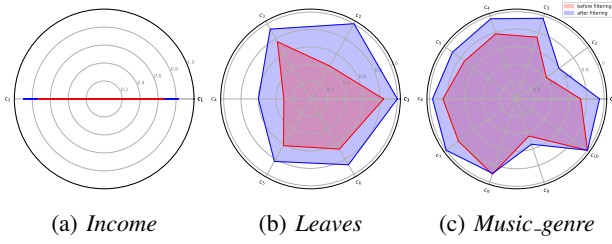

(a) *Income*     (b) *Leaves*     (c) *Music_genre*

*Figure 4.* Comparison of aggregation accuracy in $\boldsymbol{X}$ and $\boldsymbol{X}_S$.

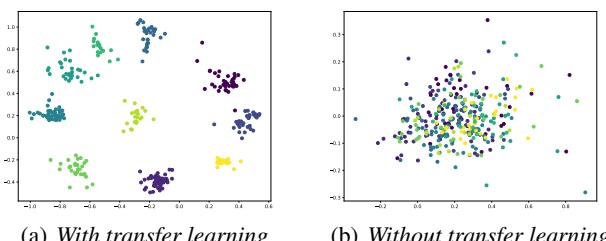

(a) *With transfer learning*     (b) *Without transfer learning*

*Figure 5.* Visualization of new embeddings learned by the Siamese network $f_T^r$ with and without transfer learning.

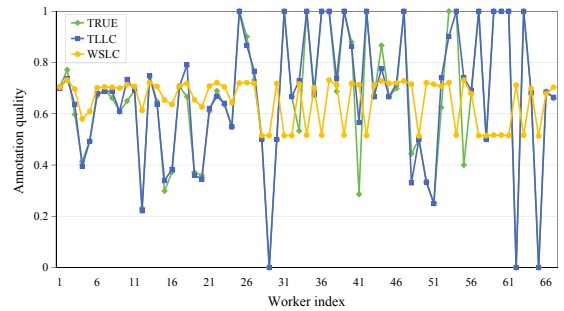

*Figure 6.* Changes in annotation quality of workers before and after label completion on dataset *Income*.

*Leaves*, TLLC consistently outperforms WSLC in aggregation accuracy across all label aggregation methods. Even on dataset *Music_genre*, significance tests indicate that TLLC still demonstrates strong potential. In this subsection, we provide a deeper analysis of TLLC, validating its underlying rationality and exploring the reasons behind its suboptimal performance on dataset *Music_genre*.

**Rationality.** To improve the performance of label completion, we introduce several innovative strategies for TLLC. First of all, when constructing $\boldsymbol{X}_S$, considering the noise in the label matrix, we draw inspiration from confident learning and design Equation (4) to filter out high-confidence annotated instances based on the initial aggregated labels. To validate the effectiveness of this strategy, we compare the aggregation accuracies in $\boldsymbol{X}$ (before filtering) and $\boldsymbol{X}_S$ (after filtering) for each class across three datasets. The detailed results are shown in Figure 4. Based on Figure 4, we can find

that after filtering, the aggregation accuracies for almost all classes across all datasets are significantly improved, which strongly supports the rationality of Equation (4).

Subsequently, considering the impact of insufficient worker modeling on label completion, we introduce transfer learning into worker modeling. To validate the rationality of this strategy, we focus on a worker with relatively few labels (r = 2) from dataset *Music_genre*. Figure 5 illustrates the new embeddings of $\boldsymbol{X}$ learned by the Siamese network $f_T^r$ corresponding to this worker, obtained through two approaches: using transfer learning (pre-training $f_S^r$ and then transferring to $f_T^r$) and without transfer learning (directly training $f_T^r$). The visualization in Figure 5 demonstrates that the former approach better clusters instances with the same true labels, indicating its ability to capture more essential knowledge for worker modeling effectively.

Finally, to complete the missing labels, we design Equation (19), which determines whether or not to complete missing labels based on the distances between the new embeddings of annotated and unannotated instances. To validate the rationality of Equation (19), we analyze the changes in annotation quality of workers before and after label completion on dataset *Income*, as shown in Figure 6 (results of datasets *Leaves* and *Music_genre* are provided in **Appendix C** due to the limited pages). From Figure 6 we can see that after label completion using WSLC, workers' annotation quality tends to converge, indicating WSLC assigns similar labels across workers. This erases workers' unique characteristics, violating **Definition 3.2**. In contrast, TLLC maintains smaller changes in workers' annotation quality, preserving their unique characteristics and better adhering to **Definition 3.2**. These results strongly validate the rationality of label completion in TLLC and confirm that $f_T^r$ effectively captures the unique attributes of $u_r$.

**Ablation experiment.** Through rationality analyses, we have preliminarily validated the effectiveness of each strategy in TLLC. To further investigate the impact of different strategies on TLLC's performance, we conduct an ablation experiment on dataset *Income*. Specifically, we fix MV as the label aggregation method to evaluate the aggregation accuracy achieved by TLLC and its variants. For clarity, we denote the variants of TLLC without instance filtering, pretraining, and transfer learning as "TLLC-IF", "TLLC-PT", and "TLLC-TL", respectively. The detailed experimental results are shown in Figure 7. Based on these results, it can be found that the performance degrades when any of these strategies is removed from TLLC. These findings further highlight the critical role of instance filtering, pretraining, and transfer learning in enhancing TLLC's performance.

**Sensitivity analysis.** In addition to evaluating the effectiveness and rationality of TLLC, we also perform a param-

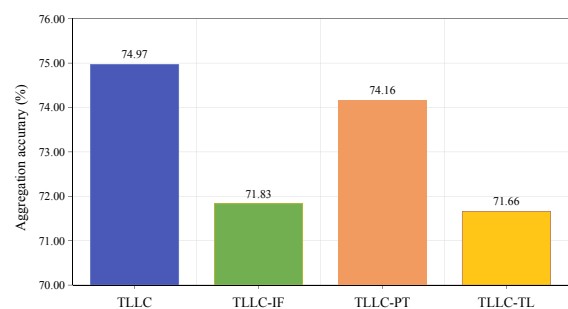

*Figure 7.* Aggregation accuracy (%) achieved by TLLC and its variants on dataset *Income*.

*Table 3.* Aggregation accuracy (%) achieved by TLLC on dataset *Income* as the parameters change.

| | New embedding dimension | | | | |
|---|---|---|---|---|---|
| Value | 2 | 4 | 6 | 8 | 10 |
| Accurary (%) | **74.94** | 71.83 | 71.66 | 73.33 | 72.66 |
| | Epochs | | | | |
| Value | 2 | 4 | 6 | 8 | 10 |
| Accurary (%) | **74.94** | 72.33 | 73.00 | 72.16 | 72.83 |
| | Batch size | | | | |
| Value | 8 | 16 | 32 | 64 | 128 |
| Accurary (%) | 71.83 | 72.50 | **74.94** | 73.33 | 73.16 |

eter sensitivity analysis for it. TLLC includes three key adjustable parameters: the new embedding dimension, the number of epochs, and the batch size. To observe the impact of these parameters on TLLC's performance, we conduct sensitivity analysis experiments on dataset *Income* (using MV as the label aggregation method). In each experiment, two parameters are fixed, while the remaining one is varied. The detailed experimental results are shown in Table 3. From these results, it can be found that TLLC's effectiveness shows only slight variation with changes in parameter values. Given that the aggregation accuracy of MV after label completion using WSLC is 71.17% (as shown in Figure 2(a)), it is clear that TLLC consistently achieves superior performance. Therefore, the effectiveness of TLLC is not highly sensitive to parameter settings.

Besides parameter settings, we conduct another set of experiments to observe the impact of datasets' missing rate (proportion of missing labels) on TLLC. The results reveal that TLLC is more effective in scenarios with a high missing rate. This finding aligns with our objective of addressing the challenges posed by insufficient worker modeling. Due to the limited pages, more detailed settings and results of these experiments are provided in **Appendix D**.

**Abnormality.** According to Figure 2(c), we can observe that the performance of TLLC on dataset *Music_genre* is less

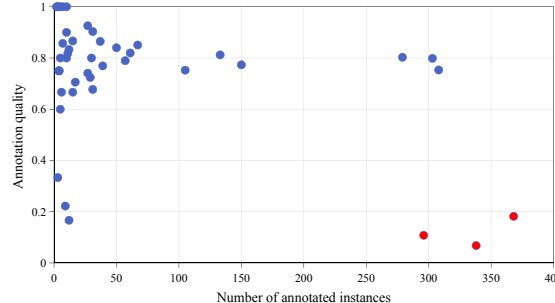

*Figure 8.* Relationship between the number of annotated instances and annotation quality for each worker in dataset *Music_genre*.

pronounced compared to its performance on datasets *Income* and *Leaves*. To explore the underlying reasons, Figure 8 illustrates the relationship between the number of annotated instances and annotation quality for each worker in dataset *Music_genre* (results of datasets *Income* and *Leaves* are provided in **Appendix E** due to the limited pages). Notably, Figure 8 identifies three adversarial workers (highlighted in red) who annotated a large number of instances with exceptionally low quality. Since these adversarial workers annotate a large number of instances, they significantly influence $f_T^r$ in TLLC during transfer learning, as $f_T^r$ captures their unique but erroneous characteristics. Furthermore, since TLLC adheres to Definition 3.2 and seeks to complete the labels workers are most likely to annotate, it inadvertently completes incorrect labels for these workers. In contrast, as shown in Figure 6, WSLC reduces the impact of adversarial workers by changing the annotation quality of workers, though this comes at the cost of erasing their unique characteristics. These observations explain the anomaly in Figure 2(c) and reveal the lack of robustness in TLLC against adversarial workers with numerous labels.

## 5. Conclusion

This paper is the first to reveal the limitations of insufficient worker modeling on label completion. To address this issue, we design a novel algorithm to construct the source and target domains from crowdsourced data, which makes it possible to introduce transfer learning into crowdsourcing. Subsequently, we train Siamese networks to model workers through transfer learning, which significantly mitigates the impact of insufficient worker modeling. Ultimately, both the theoretical analysis and experimental results validate the effectiveness and rationality of the TLLC we proposed.

However, the experimental results also highlight some limitations of TLLC, particularly the lack of robustness against adversarial workers who annotated a large number of instances. Refining the transfer learning process to address this issue remains a crucial direction for future research to improve the performance of TLLC.

## Acknowledgments

The work was partially supported by National Natural Science Foundation of China (62276241) and Hubei Provincial Collaborative Innovation Center for Basic Education Information Technology Services (OFHUE202312).

## Impact Statement

This paper presents work whose goal is to advance the field of Machine Learning. There are many potential societal consequences of our work, none which we feel must be specifically highlighted here.

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

## A. Summary of commonly used notations.

As the method proposed in this paper involves crowdsourcing, worker modeling, transfer learning, and Siamese networks, a large number of notations are introduced. Therefore, Table 4 is provided to summarize the notations used in the paper to reduce reading difficulty.

*Table 4.* Summary of commonly used notations.

| Notation | Description | Notation | Description |
|---|---|---|---|
| $D$ | data | $\boldsymbol{x}$ | instance |
| $N$ | number of instances in $D$ | $x$ | attribute value |
| $M$ | dimension of attributes | $\boldsymbol{z}$ | new embedding |
| $R$ | number of workers | $z$ | embedding value |
| $Q$ | number of classes | $y$ | true label |
| $A$ | attribute | $\hat{y}$ | aggregated label |
| $S$ | source | $y'$ | supervisory information |
| $T$ | target | $l$ | noisy label |
| $K$ | dimension of the new embedding | $l = -1$ | missing label |
| $O$ | time complexity | $\hat{l}$ | completed label |
| $\hat{D}$ | completed data | $f$ | objective predictive function |
| $D^r$ | data corresponding to $u_r$ | $f_d$ | distance function in $f$ |
| $D'$ | training data | $f_g$ | embedding function in $f$ |
| $L^1$ | $L^1$ divergence | $f^r$ | function corresponding to $u_r$ |
| $\boldsymbol{X}$ | all instances in $D$ | $\epsilon$ | error |
| $\boldsymbol{X}^r$ | instances annotated by $u_r$ | $\lambda$ | difference of functions |
| $\bar{\boldsymbol{X}}^r$ | instances not annotated by $u_r$ | $d$ | distance |
| $\boldsymbol{L}$ | multiple noisy label set | $c$ | class |
| $\boldsymbol{L}^r$ | labels $u_r$ annotated for $\boldsymbol{X}^r$ | $u$ | worker |
| $P(\boldsymbol{X})$ | marginal probability distribution | $\sigma^2$ | variance |
| $P(c \mid \boldsymbol{L})$ | probability / confidence | $\mu$ | average value |
| $\mathcal{D}$ | domain | $\boldsymbol{z}^r$ | new embedding corresponding to $f^r$ |
| $\mathcal{T}$ | task | $\bar{\boldsymbol{z}}_q$ | centroid corresponding to $c_q$ |
| $\mathcal{X}$ | attribute space | $d^r$ | distance corresponding to $f^r$ |
| $\mathcal{Y}$ | label space | $d_q$ | distance corresponding to $c_q$ |
| $\mathcal{N}$ | Gaussian distribution | $\mid \bullet \mid$ | set size |
| $\mathcal{L}$ | loss function | $\widetilde{\bullet}$ | example object |
| $\mathbb{E}$ | expectation | $\delta(\bullet)$ | indicator function |

## B. Detailed information of experimental datasets.

The descriptions of three real-world crowdsourced datasets are listed in Table 5. Here, "#Instances" denotes the number of instances, "#Workers" denotes the number of workers, "#Labels" denotes the number of labels, "#Attributes" denotes the number of attributes, and "#Classes" denotes the number of classes. These datasets are collected from different application scenarios and represent different crowdsourcing requirements. We have uploaded these datasets and our codes, which are available at https://github.com/jiangliangxiao/TLLC.

## C. Changes in annotation quality of workers on datasets *Leaves* and *Music_genre*.

The changes in annotation quality of workers before and after label completion on datasets *Leaves* and *Music_genre* are shown in Figure 9. Similar to Figure 6, it can be found from Figure 9 that after label completion using WSLC, workers' annotation quality tends to converge, indicating WSLC assigns similar labels across workers. This erases workers' unique characteristics, violating **Definition 3.2**. In contrast, TLLC maintains smaller changes in workers' annotation quality, preserving their unique characteristics and better adhering to **Definition 3.2**. These results strongly validate the rationality

*Table 5.* Descriptions of the three real-world datasets used in our experiments.

| Dataset | *Income* | *Leaves* | *Music_genre* |
|---|---|---|---|
| #Instances | 600 | 384 | 700 |
| #Workers | 67 | 83 | 44 |
| #Labels | 6000 | 3840 | 2946 |
| #Attributes | 10 (nominal) | 64 (numeric) | 31 (numeric) |
| #Classes | 2 | 6 | 10 |
| Averaged #Labels per instance | 10 | 10 | 4.2 |
| Proportion of missing labels | 0.85 | 0.88 | 0.90 |

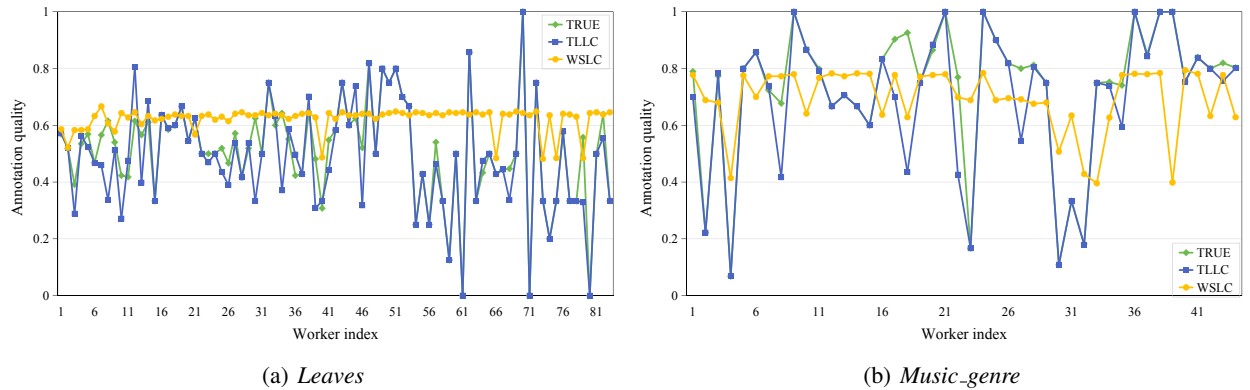

(a) *Leaves*

(b) *Music_genre*

*Figure 9.* Changes in annotation quality of workers before and after label completion on datasets *Leaves* and *Music_genre*.

*Table 6.* Aggregation accuracy (%) achieved by WSLC and TLLC as the missing rate changes.

| Missing rate | 0.9 | 0.7 | 0.5 | 0.3 | 0.1 |
|---|---|---|---|---|---|
| WSLC | 70.17 | 80.33 | 81.67 | **92.67** | **94.83** |
| TLLC | **71.16** | **81.16** | **82.33** | 92.33 | 94.00 |

of label completion in TLLC and confirm that $f_T^r$ effectively captures the unique attributes of $u_r$.

## D. More detailed settings and results of sensitivity analysis experiments.

To analyze the impact of the missing rate on the performance of TLLC, we conduct simulated experiments on dataset *Income*. Specifically, we simulate 40 workers annotating the dataset, where each worker's annotation quality is randomly generated from a uniform distribution of [0.55, 0.75]. The missing rate is controlled by adjusting workers' annotation probabilities, ensuring it varies from 0.9 to 0.1 in intervals of 0.2. When the label aggregation method is fixed as MV, the aggregation accuracy achieved by WSLC and TLLC is shown in Table 6. It can be found that when the missing rate exceeds 0.5, TLLC outperforms WSLC. However, as the missing rate decreases further, WSLC becomes more effective than TLLC. Therefore, TLLC is more effective in scenarios with a high missing rate. This finding aligns with our objective of addressing the challenges posed by insufficient worker modeling. TLLC improves label completion by addressing insufficient worker modeling. A higher missing rate increases the likelihood of insufficient modeling, making TLLC's advantages more pronounced. As the missing rate decreases, TLLC's effectiveness relative to WSLC gradually diminishes.

## E. Relationship distribution in datasets *Income* and *Leaves*.

The relationships between the number of annotated instances and annotation quality for each worker in datasets *Income* and *Leaves* are shown in Figure 10. From Figure 10, it can be found that datasets *Income* and *Leaves* also contain adversarial workers with low annotation quality. However, since they did not annotate a large number of instances, TLLC's performance

is insensitive to them. This finding indicates that TLLC lacks robustness only against adversarial workers who annotated a large number of instances.

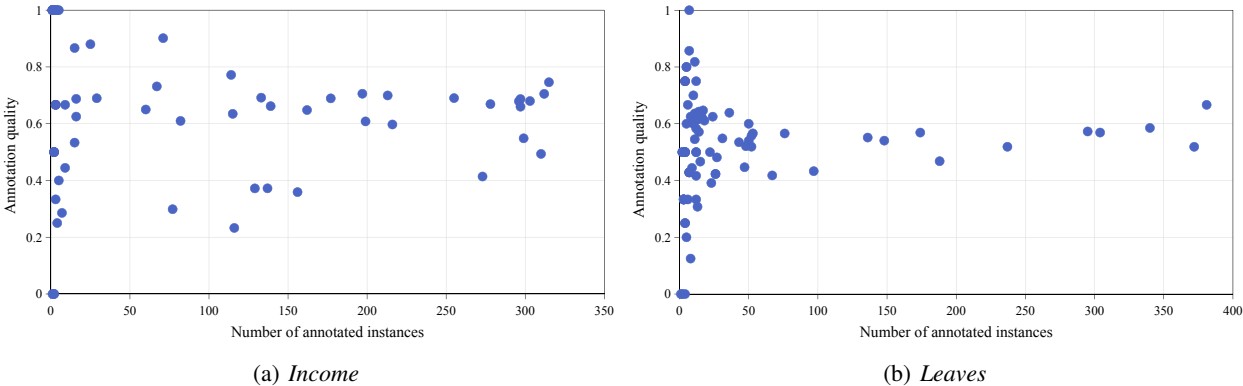

(a) *Income*                                      (b) *Leaves*

*Figure 10.* Relationship between the number of annotated instances and annotation quality for each worker in datasets *Income* and *Leaves*.

