# OpenReview forum: "TLLC: Transfer Learning-based Label Completion for Crowdsourcing"
_ICML.cc/2025/Conference — ICML 2025 spotlightposter_

### Official Review · Reviewer_3DPZ · 2025-02-24

**Overall Recommendation:** 4

**Summary:**

To complete the missing labels, this paper proposes a novel label completion method for crowdsourcing by utilizing transfer learning. All high-confidence instances from the original data are selected as the source domain, and a Siamese network is pretrained based on the instances coming from the source domain. After transferring the pretrained network to the target domain, some fine-tunings are applied to obtain the unique characteristics of each annotator, also called worker modeling. Corresponding theorems are provided to prove that the proposed transfer learning-based method can reduce the generalization error. Experimental results and related analysis also demonstrate the effectiveness of the proposal.

## update after rebuttal
The author's response satisfactorily resolves my concerns, and upon considering the feedback from the other reviewers, I support the acceptance of this paper. Thus, I keep my initial rating unchanged.

**Claims And Evidence:**

Yes.

**Essential References Not Discussed:**

No.

**Experimental Designs Or Analyses:**

Yes. This paper designs the related experiments to show the effectiveness of the proposal, and gives some analysis to explain the advantages or disadvantages on different situations.

**Methods And Evaluation Criteria:**

Yes.

**Other Comments Or Suggestions:**

1 In Line 211, ‘we’ should be ‘We’.
2 In Equation (3), it would be better to change the representation of the average value P, since the overline has been used to denote the complement set.

**Other Strengths And Weaknesses:**

Strengths:
1 This paper studies the label completion problem in crowdsourcing area, which is a very important issue as usually the missing proportion is very high in real-world scenarios.
2 This paper proposes a label completion method based on transfer learning. The comparative experiments demonstrate that the idea is simple yet efficient.
3 This paper theoretically discusses the generalization error reduction by utilizing transfer learning into label completion problem.
4 This paper constructs a novel source domain algorithm, which can be easily extended to various application scenarios. The theoretical analysis guarantees its effectiveness.

Weaknesses:
1 There appears to be an error in Equation (11). In the MSE loss, there should be a minus sign ‘-’ instead of a comma ‘,’. Additionally, the notation for $x_{i1}$ and $x_{i2}$ is somewhat confusing to me, as it could easily be interpreted as the values of two attributes for the same instance $x_i$.
2 Some parts of the description are not clear enough. For example, what is the ‘g’ of the time complexity O(g) refer to? If g is not a kind of quantity, O(N^2g) is not a correct expression.
3 In the appendix, it is shown that the missing rates for all three datasets are very high, exceeding 0.85. Do the different missing rates have an impact on the method proposed in this paper? What kind of impact do they have?

**Questions For Authors:**

1 Some parts of the description are not clear enough. For example, what is the ‘g’ of the time complexity O(g) refer to? If g is not a kind of quantity, O(N^2g) is not a correct expression.
2 In the appendix, it is shown that the missing rates for all three datasets are very high, exceeding 0.85. Do the different missing rates have an impact on the method proposed in this paper? What kind of impact do they have?

**Relation To Broader Scientific Literature:**

This paper discusses the effectiveness of transfer learning for crowdsourcing problem, and proposes a method of how to construct source domain data, which can theoretically guarantee the reduction of generalization error. The source domain construction method and theoretical analysis can be easily applied to other transfer learning-based application areas.

**Theoretical Claims:**

Yes. The theoretical claims are all about reducing the generalization error, and the proofs are correct.

---

> ### Author Rebuttal · Authors · 2025-03-31
>
> **Q1:** There appears to be an error in Equation (11). In the MSE loss, there should be a minus sign ‘-’ instead of a comma ‘,’. Additionally, the notation for $x_{i1}$ and $x_{i2}$ is somewhat confusing to me, as it could easily be interpreted as the values of two attributes for the same instance $x_{i}$.
>
> **Author Response to Q1:** Thanks for your valuable comments. In the MSE loss, it should indeed be a minus sign instead of a comma. Meanwhile, to address the reviewer’s concerns regarding notation, we will revise $x\_{i1}$ and $x\_{i2}$ to $x\_{i}^1$ and $x\_{i}^2$. Accordingly, in the final version of the paper, we will update Equation (11) as follows:
> $$
> \mathcal{L}\_{mse} = \frac{1}{|\widetilde{\mathbf{X}}|^2}\sum\_{i=1}^{|\widetilde{\mathbf{X}}|^2}(\widetilde{f}\_{d}(\widetilde{f}\_{g}(\mathbf{x}\_{i}^1), \widetilde{f}\_{g}(\mathbf{x}\_{i}^2)) - y\_i')^2.
> $$
> Thanks again for your valuable comments.
>
> **Q2:** Some parts of the description are not clear enough. For example, what is the ‘g’ of the time complexity O(g) refer to? If g is not a kind of quantity, O(N^2g) is not a correct expression.
>
> **Author Response to Q2:** Thanks for your valuable comments. In our paper, $\widetilde{f}\_{g}$ is explicitly defined as the network structure used for learning new embeddings in a Siamese network. The time complexity of $\widetilde{f}\_{g}$ is not determined by Algorithm 2 but is instead related to the scale of $\widetilde{f}\_{g}$. Therefore, we denote its time complexity as $O(g)$. In the final version of the paper, we will explicitly clarify that $g$ represents the scale of $\widetilde{f}\_{g}$. Additionally, we will carefully review and refine other descriptions to improve clarity and precision. Thanks again for your valuable comments.
>
> **Q3:** In the appendix, it is shown that the missing rates for all three datasets are very high, exceeding 0.85. Do the different missing rates have an impact on the method proposed in this paper? What kind of impact do they have?
>
> **Author Response to Q3:** Thanks for your valuable comments. The missing rates significantly impact TLLC’s performance. Specifically, TLLC improves label completion by addressing insufficient worker modeling. A higher missing rate increases the likelihood of insufficient modeling, making TLLC’s advantages more pronounced. Conversely, as the missing rate decreases, TLLC’s effectiveness relative to WSLC gradually diminishes. To validate this analysis, we conduct simulated experiments on the Income dataset. We simulate 40 workers annotating the dataset, where each worker’s annotation quality is randomly generated from a uniform distribution of [0.55, 0.75]. The missing rate is controlled by adjusting workers’ annotation probabilities, ensuring it varies from 0.9 to 0.1 in intervals of 0.2. When the label aggregation algorithm is fixed as MV, the label completion performance of WSLC and TLLC is as follows:
> |Missing Rates|0.9|0.7|0.5|0.3|0.1|
> |--|--|--|--|--|--|
> |WSLC|70.17%|80.33%|81.67%|**92.67%**|**94.83%**|
> |TLLC|**71.16%**|**81.16%**|**82.33%**|92.33%|94.00%|
> ||
>
> These results confirm our analysis: when the missing rate exceeds 0.5, TLLC outperforms WSLC. However, as the missing rate decreases further, WSLC becomes more effective than TLLC. In the final version of the paper, we will thoroughly describe and discuss these experiments about missing rates. Thanks again for your valuable comments.
>
> **Q4:** In Line 211, ‘we’ should be ‘We’. In Equation (3), it would be better to change the representation of the average value P, since the overline has been used to denote the complement set.
>
> **Author Response to Q4:** Thanks for your valuable comments. In the final version of the paper, we will revise the statement in line 211 from "we set $y_{ij}'$ to 0 if $l_{i} = l_{j}$" to "We set $y_{ij}'$ to 0 if $l_{i} = l_{j}$". Additionally, to avoid ambiguity caused by using the overline to represent both the average value and the complement set, we will consistently use $\mu$ to denote the average value. Accordingly, Equation (3) will be revised as follows:
> $$
>  \mu\_{c\_q} = \frac{\sum\_{i=1}^{N}\delta(\hat{y}\_i, c\_q)P(\hat{y}\_i|\mathbf{L}\_i)}{\sum\_{i=1}^{N}\delta(\hat{y}\_i, c\_q)}.
> $$
> Equations (4) and Table 2 will also be updated accordingly. Thanks again for your valuable comments.

---

### Official Review · Reviewer_1QrC · 2025-03-06

**Overall Recommendation:** 4

**Summary:**

Existing worker modeling-based label completion methods have successfully improved the performance of label completion, but they remain constrained by the insufficient annotated instances per worker. To address this issue, this paper proposes a transfer learning-based label completion (TLLC) method. TLLC begins by identifying all high-confidence instances from the whole crowdsourced data as a source domain to pretrain a Siamese network. Next, TLLC transfers the pretrained network to target domains, where it is fine-tuned using the instances annotated by each worker individually. Finally, TLLC utilizes the new embeddings learned by the transferred network to complete the missing labels for each worker. Experimental results validate the effectiveness and rationality of TLLC.

**Claims And Evidence:**

There are three important claims made in the paper:
1) Worker modeling has been proved to be a powerful strategy to improve the performance of label completion.
2) Workers typically annotate only a few instances, which leads to insufficient worker modeling and thus limiting the improvement of label completion.
3) The proposed transfer learning-based label completion method helps alleviate the issue of insufficient worker modeling.
In response to these claims, this paper provides the corresponding evidence as follows:
1) For Claim 1, this paper summarizes and discusses existing label completion methods in the introduction and related work sections. The latest label completion methods have indeed achieved impressive results by leveraging worker modeling.
2) For Claim 2, the paper cites an existing work (Jung & Lease, 2012) to emphasize that, in real-world scenarios, each worker typically annotates only a few instances. Furthermore, the description of real-world datasets in Section 4.1 similarly supports this phenomenon. Based on this phenomenon, existing worker modeling-based label completion methods are indeed constrained by the insufficient annotated instances per worker.
3) For Claim 3, the experimental results presented in Section 4 demonstrate the effectiveness of the proposed method. Specifically, Figures 1 and 2 validate the effectiveness of TLLC for improving the performance of label completion, while Figure 4 independently validates the effectiveness of transfer learning for insufficient worker modeling.

**Essential References Not Discussed:**

No. All essential references have been cited/discussed in the paper.

**Experimental Designs Or Analyses:**

Yes. The experimental section first validates the effectiveness of TLLC through comparative experiments and significance tests. Then, the rationality of TLLC is verified through ablation studies on each strategy. Finally, potential limitations of TLLC are analyzed by discussing its abnormality.

**Methods And Evaluation Criteria:**

Yes. In the proposed method, using transfer learning to address the issue of insufficient worker modeling is reasonable. Regarding evaluation criteria, this paper adopts the aggregation accuracy, which is commonly used in other label completion studies.

**Other Comments Or Suggestions:**

I have found few typos as follows:
1) In Equation 11, there should be a minus sign before ${y}_i’$ instead of a comma.
2) On line 366 in page 7, “dataset Music_genre dataset” should be “dataset Music_genre”.

**Other Strengths And Weaknesses:**

Strengths:
1) This paper reveals a critical limitation of existing label completion methods: insufficient worker modeling due to insufficient annotated instances per worker. By introducing transfer learning, the proposed method effectively addresses this issue.
2) The use of a Siamese network for both pretraining and fine-tuning is innovative in worker modeling. The idea of constructing source and target domains from the same crowdsourced data and leveraging high-confidence instances for pretraining adds robustness to the method.
3) The paper provides theoretical proofs to support the claims of the paper (Theorems 3.6, 3.7, and 3.8). These theorems and proofs strengthen the claims about reduced generalization error and robustness against noise.
4) This paper provides extensive experiments to validate the effectiveness and rationality of TLLC. The paper first validates the effectiveness of TLLC through comparative experiments and significance tests. Then, the rationality of TLLC is validated through ablation studies on each strategy. Finally, potential limitations of TLLC are analyzed by discussing its abnormality.

Weaknesses:
1) This paper uses three algorithms to describe the construction of source and target domains, worker modeling, and label completion, respectively. However, how these three algorithms are combined to form the complete TLLC remains unclear. A framework diagram is needed to provide a complete introduction of TLLC.
2) The current ablation study is not comprehensive. Although the paper validates the rationality of each strategy in TLLC from multiple perspectives, this is not directly reflected in the evaluation criteria of label completion. In my opinion, it is necessary to construct a complete ablation study based on aggregation accuracy.

**Questions For Authors:**

See the Weaknesses parts：
1) This paper uses three algorithms to describe the construction of source and target domains, worker modeling, and label completion, respectively. However, how these three algorithms are combined to form the complete TLLC remains unclear. A framework diagram is needed to provide a complete introduction of TLLC.
2) The current ablation study is not comprehensive. Although the paper validates the rationality of each strategy in TLLC from multiple perspectives, this is not directly reflected in the evaluation criteria of label completion. In my opinion, it is necessary to construct a complete ablation study based on aggregation accuracy.

**Relation To Broader Scientific Literature:**

This paper is the first work to introduce transfer learning into label completion, addressing the impact of insufficient worker modeling on label completion, thereby further improving the performance of label completion.

**Theoretical Claims:**

Yes. The theories and corresponding proofs presented in the paper support the effectiveness and rationality of the proposed method.

---

> ### Author Rebuttal · Authors · 2025-03-31
>
> **Q1:** This paper uses three algorithms to describe the construction of source and target domains, worker modeling, and label completion, respectively. However, how these three algorithms are combined to form the complete TLLC remains unclear. A framework diagram is needed to provide a complete introduction of TLLC.
>
> **Author Response to Q1:** Thanks for your valuable comments. The complete process of TLLC is as follows: Given a crowdsourced dataset, we first construct the source and target domains. Next, we pretrain a Siamese network on the source domain. Subsequently, for each worker's corresponding target domain, we individually transfer the pretrained network. Finally, we use the transferred network to learn new embeddings for each worker and infer the missing labels based on these embeddings. In essence, this process corresponds to the sequential execution of Algorithm 1, Algorithm 2, and Algorithm 3. In the final version of the paper, according to the reviewer’s comments, we will incorporate a framework diagram to clearly present the complete process of TLLC. Thanks again for your valuable comments.
>
> **Q2:** The current ablation study is not comprehensive. Although the paper validates the rationality of each strategy in TLLC from multiple perspectives, this is not directly reflected in the evaluation criteria of label completion. In my opinion, it is necessary to construct a complete ablation study based on aggregation accuracy.
>
> **Author Response to Q2:** Thanks for your valuable comments. To address the reviewer’s concerns, we conduct an ablation study on the Income dataset for TLLC and its variants (using MV as the label aggregation method). The experimental results are as follows:
> | |TLLC|TLLC1|TLLC2|TLLC3|
> |--|--|--|--|--|
> |Aggregation Accuracy|**74.97%**|71.83%|74.16%|71.66%|
> ||
>
> Here, TLLC1, TLLC2, and TLLC3 represent the variants of TLLC without instance filtering, pretraining, and transfer training, respectively. Considering that the aggregation accuracy of MV before completion is 71.17% (as shown in Figure 1), it can be observed that all TLLC variants outperform MV. Meanwhile, each variant performs worse than the complete TLLC, further indicating the superior performance and rationality of TLLC. In the final version of the paper, we will provide a detailed description and discussion of the setup and results of this ablation study. Thanks again for your valuable comments.
>
> **Q3:** In Equation 11, there should be a minus sign before $y_i’$ instead of a comma.
>
> **Author Response to Q3:** Thanks for your valuable comments. In the MSE loss, it should indeed be a minus sign instead of a comma. Meanwhile, to address the reviewer **3DPZ’s** concerns regarding notation, we will revise $x\_{i1}$ and $x\_{i2}$ to $x\_{i}^1$ and $x\_{i}^2$. Accordingly, in the final version of the paper, we will update Equation (11) as follows:
> $$
> \mathcal{L}\_{mse} = \frac{1}{|\widetilde{\mathbf{X}}|^2}\sum\_{i=1}^{|\widetilde{\mathbf{X}}|^2}(\widetilde{f}\_{d}(\widetilde{f}\_{g}(\mathbf{x}\_{i}^1), \widetilde{f}\_{g}(\mathbf{x}\_{i}^2)) - y\_i')^2.
> $$
> Thanks again for your valuable comments.
>
> **Q4:** On line 366 in page 7, “dataset Music_genre dataset” should be “dataset Music_genre”.
>
> **Author Response to Q4:** Thanks for your valuable comments. In the final version of the paper, we will correct "dataset Music_genre dataset" to "dataset Music_genre" on line 366 in page 7. Additionally, we will double-check and improve the writing of our paper. Thanks again for your valuable comments.

---

> > ### Comment · Reviewer_1QrC · 2025-04-07
> >
> > Thank you for the author's response. After considering the other reviewers' comments, I have decided to maintain my original rating.

---

### Official Review · Reviewer_ksp6 · 2025-03-09

**Overall Recommendation:** 4

**Summary:**

This paper at first reveals the limitations of existing methods that leverage worker modeling to improve label completion for Crowdsourcing and then proposes a novel transfer learning-based label completion (TLLC) method, which introduces transfer learning to avoid insufficient worker modeling and leverages the new embeddings learned by the transferred network to complete missing labels.

**Claims And Evidence:**

Yes

**Essential References Not Discussed:**

NO

**Experimental Designs Or Analyses:**

Yes

**Methods And Evaluation Criteria:**

Yes

**Other Comments Or Suggestions:**

NO

**Other Strengths And Weaknesses:**

Strengths:
1. The authors reveal the limitations of existing methods that leverage worker modeling to improve label completion for Crowdsourcing.
2. To address this issue, the authors propose a novel transfer learning-based label completion (TLLC) method, which is the first work to introduce transfer learning to avoid insufficient worker modeling and then leverages the new embeddings learned by the transferred network to complete missing labels.
3. The authors conduct extensive experiments to validate the effectiveness, rationality and abnormality of the proposed TLLC on the widely used real-world datasets.
4. The organization of the paper is quite good and it is easy to follow the topic and the proposed method.

Weaknesses:
1. The proposed TLLC transfers the pretrained Siamese network to the target domain. In the paper, the authors just said: “Specifically, we set up both $f_S$ and $f_T$ as Siamese networks with the same structure (Li et al., 2022).” What are the detailed network structure and parameter settings?
2. Although the authors have already provided a deeper analysis of TLLC to validate its underlying rationality: 1) With and without instance filtering; 2) With and without transfer learning, a group of thorough ablation experiments are needed.
3. On page 8, Figure 6 illustrates the relationship between the number of annotated instances and annotation quality for each worker in dataset Music genre. Why are the axes named Aggregation accuracy (%) and Number of aggregated instances?

**Questions For Authors:**

The proposed TLLC transfers the pretrained Siamese network to the target domain. In the paper, the authors just said: “Specifically, we set up both $f_S$ and $f_T$ as Siamese networks with the same structure (Li et al., 2022).” What are the detailed network structure and parameter settings?
    Although the authors have already provided a deeper analysis of TLLC to validate its underlying rationality: 1) With and without instance filtering; 2) With and without transfer learning, a group of thorough ablation experiments are needed.
    On page 8, Figure 6 illustrates the relationship between the number of annotated instances and annotation quality for each worker in dataset Music genre. Why are the axes named Aggregation accuracy (%) and Number of aggregated instances?

**Relation To Broader Scientific Literature:**

This paper proposes a novel transfer learning-based label completion (TLLC) method, which is the first work to introduce transfer learning to avoid insufficient worker modeling and then leverages the new embeddings learned by the transferred network to complete missing labels.

**Theoretical Claims:**

Yes

---

> ### Author Rebuttal · Authors · 2025-03-31
>
> **Q1:** The proposed TLLC transfers the pretrained Siamese network to the target domain. In the paper, the authors just said: “Specifically, we set up both and as Siamese networks with the same structure (Li et al., 2022).” What are the detailed network structure and parameter settings?
>
> **Author Response to Q1:** Thanks for your valuable comments. In TLLC, the Siamese network is used to model each worker, and the new embeddings it learns are ultimately used to complete the worker’s missing labels. Since each worker annotates only a few instances, we set the network to a small scale to ensure convergence. The detailed network structure and parameter settings are as follows:
> |Layer Type|Output Dimension|Activation Function|
> |--|:--:|:--:|
> |Input Layer|128|ReLU|
> |Fully Connected Layer|64|ReLU|
> |Output Layer|2|-|
> ||
>
> To address the reviewer’s concerns, we will include the above information in the final version of the paper. Additionally, we have already submitted our code and datasets in Supplementary Material. At the same time, we will also open-source our code to facilitate the reproduction of our results once our paper is accepted. Thanks again for your valuable comments.
>
> **Q2:** Although the authors have already provided a deeper analysis of TLLC to validate its underlying rationality: 1) With and without instance filtering; 2) With and without transfer learning, a group of thorough ablation experiments are needed.
>
> **Author Response to Q2:** Thanks for your valuable comments. To address the reviewer’s concerns, we conduct an ablation study on the Income dataset for TLLC and its variants (using MV as the label aggregation method). The experimental results are as follows:
> | |TLLC|TLLC1|TLLC2|TLLC3|
> |--|--|--|--|--|
> |Aggregation Accuracy|**74.97%**|71.83%|74.16%|71.66%|
> ||
>
> Here, TLLC1, TLLC2, and TLLC3 represent the variants of TLLC without instance filtering, pretraining, and transfer training, respectively. Considering that the aggregation accuracy of MV before completion is 71.17% (as shown in Figure 1), it can be observed that all TLLC variants outperform MV. Meanwhile, each variant performs worse than the complete TLLC, further indicating the superior performance and rationality of TLLC. In the final version of the paper, we will provide a detailed description and discussion of the setup and results of this ablation study. Thanks again for your valuable comments.
>
> **Q3:** On page 8, Figure 6 illustrates the relationship between the number of annotated instances and annotation quality for each worker in dataset Music genre. Why are the axes named Aggregation accuracy (%) and Number of aggregated instances?
>
> **Author Response to Q3:** Thanks for your valuable comments, and we apologize for our typos in Figure 6. In the final version of the paper, we will revise the titles of the horizontal and vertical axes in Figure 6 to "Number of annotated instances" and "Annotation accuracy (%)". Similarly, Figure 8 will be adjusted accordingly. Additionally, we will double-check and improve the writing of our paper. Thanks again for your valuable comments.

---

> > ### Comment · Reviewer_ksp6 · 2025-04-05
> >
> > Thank you for the clarifications. My concerns have been addressed. After considering other reviewers' feedback, I will maintain my positive recommendation.

---

### Official Review · Reviewer_x4PP · 2025-03-16

**Overall Recommendation:** 4

**Summary:**

The paper proposes a Transfer Learning-based Label Completion (TLLC) method for crowdsourcing scenarios. The authors address the issue of sparse label matrices, where individual workers annotate only a few instances, leading to insufficient worker modeling and poor label completion. The key idea of TLLC is to pre-train a Siamese network on high-confidence instances (source domain) and transfer it to model individual workers (target domain), thereby improving label completion. The method is evaluated against WSLC across three real-world datasets.

**Claims And Evidence:**

Most of the claims in this paper are well demonstrated. However, I still have some concerns:
1) While Theorem 3.8 suggests that TLLC is resistant to i.i.d. Gaussian noise, real-world crowdsourcing noise is often adversarial (workers deliberately provide incorrect labels). Figure 6 shows that TLLC fails to handle adversarial workers, leading to poor performance on the Music Genre dataset.
2) The paper claims that TLLC’s complexity is O(N²Rg), but lacks the theoretical and empirical comparison with WSLC.

**Essential References Not Discussed:**

N/A

**Experimental Designs Or Analyses:**

I have checked the experimental designs and analyses of this paper. Experimental design is mostly valid, with realistic datasets and strong statistical tests. However, the evaluation has notable limitations:
1) It lacks comparisons with other label completion baselines beyond WSLC, making it unclear how TLLC performs against broader alternatives.
2) The experimental analysis lacks depth in several aspects. While the paper provides basic performance comparisons and statistical tests, it does not thoroughly investigate why TLLC performs well in some cases but struggles in others.

**Methods And Evaluation Criteria:**

The use of transfer learning to improve worker modeling makes sense given the sparse crowdsourced label matrix problem. However, there are some issues that should be further resolved:
1) Only WSLC is used as a label completion baseline and more label completion baselines are needed to demonstrate to effectiveness of proposed TLLC.
2) The paper does not provide a runtime comparison between TLLC and other methods. Given the O(N²Rg) complexity, TLLC may be computationally expensive, but this is not empirically analyzed.

**Other Comments Or Suggestions:**

Please see the weakness.

**Other Strengths And Weaknesses:**

Strengths:
1) The paper provides a well-structured introduction to the problem of sparse crowdsourced label matrices and explains why worker modeling is essential for label completion.
2) The analysis of worker annotation quality before and after label completion (Figure 5) is an insightful addition.

Weaknesses:
1) The core techniques—transfer learning for label completion, worker modeling, and label filtering—are all adaptations of existing methods, rather than fundamentally new contributions.
2) Lack of benchmarks against more label completion methods. It remains unclear if TLLC truly outperforms all alternatives.
3) The paper does not analyze where TLLC makes mistakes, nor does it investigate failure cases in depth.
4) No sensitivity analysis is conducted on hyperparameters, which could impact model stability.

**Questions For Authors:**

Please refer to the concerns above.

**Relation To Broader Scientific Literature:**

TLLC builds on well-established ideas in transfer learning, worker modeling, and label completion. However, its novelty is limited because similar techniques exist in prior literature. Stronger comparisons with alternative label completion methods are needed to clarify its contribution.

**Theoretical Claims:**

The paper presents three key theoretical claims, and I have checked the logic and correctness of these proofs and identified the following issues:
The derivation of Theorem 3.8 is mathematically correct for i.i.d. Gaussian noise. However, real-world crowdsourcing noise is not i.i.d. Workers can introduce systematic bias rather than random Gaussian noise.

---

> ### Author Rebuttal · Authors · 2025-03-31
>
> Thanks a lot for your comments. Please find our detailed responses to your concerns as follows.
>
> **Author Response to Contributions:** This paper is the first to identify and address the limitation from insufficient worker modeling. Moreover, based on our review of related work, this paper is also the first to introduce transfer learning into label completion. While transfer learning is a well-established technique, its application in crowdsourcing—particularly when only a single crowdsourced dataset is available—poses a significant challenge in constructing source and target domains. Our paper proposes a novel algorithm to address this issue, achieving impressive results. In the final version of the paper, we will expand on these contributions in detail.
>
> **Author Response to Benchmarks:** Among existing label completion methods, apart from WSLC, only PG-TAC (Zhou & He, 2016) can handle both binary and multi-class classification problems (consistent with TLLC). Therefore, we compare TLLC with PG-TAC on the Income and Leaves datasets. The results are as follows:
> | |MV|GTIC|DEWSMV|MNLDP|AALI|LAGNN|
> |--|--|--|--|--|--|--|
> |PG-TAC (Income)|72.00%|71.67%|72.17%|**73.00%**|73.17%|71.83%|
> |TLLC (Income)|**74.97%**|**74.67%**|**74.80%**|72.67%|**75.37%**|**74.67%**|
> ||
> |PG-TAC (Leaves)|63.54%|62.76%|63.54%|64.32%|64.58%|63.54%|
> |TLLC (Leaves)|**68.88%**|**67.19%**|**69.01%**|**69.79%**|**72.40%**|**68.75%**|
> ||
>
> These results further demonstrate the superior performance of TLLC.
>
> **Author Response to Experimental Analysis:** In our current paper, we analyze TLLC’s rationality and abnormality in the Discussion and Analysis subsection. For rationality, we explain why TLLC performs well by analyzing the effectiveness of each component in TLLC (see the Rationality paragraph on page 7). For abnormality, we reveal that TLLC is not robust to adversarial workers providing numerous labels, and we discuss the reasons behind this phenomenon (see the Abnormality paragraph on page 8). Additionally, to address the reviewer **ksp6’s Q2**, we conduct an ablation study on the Income dataset. The results of the ablation study further indicate the rationality of TLLC. In the final version of the paper, we will provide a more in-depth analysis based on the ablation study.
>
> **Author Response to Sensitivity Analysis:** The hyperparameters in TLLC include the new embedding dimension ($K$), the number of epochs, and the batch size. We conduct parameter sensitivity analysis experiments on the Income dataset (using MV as the label aggregation method) to observe TLLC’s performance. In each experiment, we fix two hyperparameters and vary the remaining one. The results are as follows:
> |$K$|2|4|6|8|10|
> |--|:--:|:--:|:--:|:--:|:--:|
> |Income|**74.94%**|71.83%|71.66%|73.33%|72.66%|
> ||
>
> |Epochs|2($Q$)|4|6|8|10|
> |--|:--:|:--:|:--:|:--:|:--:|
> |Income|**74.94%**|72.33%|73.00%|72.16%|72.83%|
> ||
>
> |Batch Size|8|16|32|64|128|
> |--|:--:|:--:|:--:|:--:|:--:|
> |Income|71.83%|72.50%|**74.94%**|73.33%|73.16%|
> ||
>
> These results show that TLLC’s performance varies slightly with changes in hyperparameter values. Considering that the aggregation accuracy of MV before label completion is 71.17%, it is evident that TLLC’s effectiveness is not highly sensitive to hyperparameter settings. In the final version of the paper, we will provide a detailed description and discussion of these parameter sensitivity experiments.
>
> **Author Response to Theoretical Claims:** As far as we know, adversarial labels are a type of noise but not the most dominant noise. In reality, workers hired from the general public usually lack expertise, and thus the noisy labels they provide are often random, satisfying the i.i.d. assumption. Moreover, Figure 6 does not indicate that TLLC fails to handle adversarial workers, but rather that it struggles with adversarial workers who annotate a large number of labels. In the final version of the paper, we will provide more explanations regarding Figure 6 and the characteristics of noisy labels to clarify these points.
>
> **Author Response to Complexity:** We conduct a new experiment on the Income dataset to compare the runtime of WSLC and TLLC. This experiment is conducted on a Windows 10 machine with an AMD Athlon(tm) X4 860K Quad Core Processor @ 3.70GHz and 16 GB of RAM. The runtime required to complete the Income dataset for WSLC and TLLC is as follows:
> |Dataset|WSLC|TLLC|
> |--|:--:|:--:|
> |Income|0.87s|150.33s|
> ||
>
> The results show that TLLC requires more runtime compared to WSLC. However, the primary computational cost of TLLC arises from transfer learning to train Siamese networks. Both transfer learning and Siamese networks are widely adopted techniques and are not computationally expensive to use in practical scenarios. Therefore, TLLC remains efficient and applicable in real-world scenarios. In the final version of the paper, we will include a detailed explanation of the computational cost of TLLC to further clarify this point.

---

### Decision · Program_Chairs · 2025-05-01

**Decision:**

Accept (spotlight poster)

**Comment:**

This paper proposes a new method to improve label completion in crowdsourcing tasks called TLLC. Its advantages include: (1) innovative use of transfer learning by repurposing high-confidence labels within the same dataset as the source domain; (2) siamese network architecture which allows for effective learning of similarities between instances; (3) theoretical backing with proofs for robustness and reduced generalization error; and (4) strong empirical performance across multiple datasets. There also exist some potential disadvantages, including high computational cost and performance degradation as the missing label rate decreases. Overall, this is a qualified paper and I recommend acceptance.